# Pre-transition effects mediate forces of assembly between transmembrane proteins

**Shachi Katira[1†], Kranthi K Mandadapu[2,3†], Suriyanarayanan Vaikuntanathan[4†], Berend Smit[1,3,5], David Chandler[1]\***

[1]Department of Chemistry, University of California, Berkeley, Berkeley, United States; [2]Chemical Sciences Division, Lawrence Berkeley National Laboratory, Berkeley, United States; [3]Department of Chemical and Biomolecular Engineering, University of California, Berkeley, Berkeley, United States; [4]Department of Chemistry, University of Chicago, Chicago, United States; [5]Laboratory of Molecular Simulation, Institut des Sciences et Ingénierie Chimiques, Ecole Polytechnique Fédérale de Lausanne, Sion, Switzerland

**Abstract** We present a mechanism for a generic, powerful force of assembly and mobility for transmembrane proteins in lipid bilayers. This force is a pre-transition (or pre-melting) effect for the first-order transition between ordered and disordered phases in the membrane. Using large-scale molecular simulation, we show that a protein with hydrophobic thickness equal to that of the disordered phase embedded in an ordered bilayer stabilizes a microscopic order–disorder interface. The stiffness of that interface is finite. When two such proteins approach each other, they assemble because assembly reduces the net interfacial energy. Analogous to the hydrophobic effect, we refer to this phenomenon as the 'orderphobic effect'. The effect is mediated by proximity to the order–disorder phase transition and the size and hydrophobic mismatch of the protein. The strength and range of forces arising from this effect are significantly larger than those that could arise from membrane elasticity for the membranes considered.

**\*For correspondence:** chandler@berkeley.edu

[†]These authors contributed equally to this work

**Competing interests:** The author declares that no competing interests exist.

## Introduction

This paper presents implications of first-order order–disorder phase transitions in lipid bilayers. The fluid mosaic model (*Singer and Nicolson, 1972*) and the lipid raft hypothesis (*Simons and Ikonen, 1997*; *Munro, 2003*) have guided intuition on how proteins diffuse and assemble in biological membranes—ordered clusters floating in an otherwise disordered fluid membrane (*Simons and Toomre, 2000*; *Lingwood and Simons, 2010*). However, recent advances show that a significant proportion of the membrane is liquid-ordered (*Swamy et al., 2006*; *Owen et al., 2012*; *Polozov et al., 2008*), with coexistence between the liquid-ordered and disordered phases. This coexistence suggests that effects of an order–disorder transition might be at play in the assembly of proteins. This possibility is studied here by examining the effects mediated by the simplest related order–disorder transition, that between solid-ordered and liquid-disordered phases.

Specifically, with molecular simulation, we study a coarse-grained model of a hydrated one-component bilayer and proteins that are added to the membrane. The model membrane exhibits two distinct phases—a solid-ordered phase and a liquid-disordered phase—and a first-order transition between them. We find that a transmembrane protein in the ordered bilayer can induce effects that resemble pre-melting (*Lipowsky, 1982*; *1984*; *Limmer and Chandler, 2014*). In particular, within the otherwise ordered membrane phase, mesoscopic disordered domains surround proteins that

**eLife digest** The membrane that surrounds cells provides a selective barrier that allows some molecules through, but blocks the path of others. A cell's membrane is made up of two layers of molecules with oily tails, and is therefore known as a bilayer. Many proteins are dotted within and on the inner and outer surfaces of the bilayer: some act as channels that control what goes in and out of the cell, while others protrude outside the cell so that they can sense changes in the environment.

Membrane proteins can move and interact within the bilayer, and various models have emerged to try to explain this dynamic system. These models are based on the membrane having some fluidity but also having regions where there is more structure, and typically describe the proteins as ordered clusters floating in an otherwise disordered fluid membrane. However, many researchers now think some proteins that pass through both layers of the bilayer (i.e., transmembrane proteins) make membranes more ordered, with a possibly gel-like state. However, it is not clear how transmembrane proteins can move and assemble together within such a relatively rigid membrane.

To investigate this, Katira, Mandadapu, Vaikuntanathan et al. carried out computer simulations using a model of a simple bilayer membrane. This membrane can exist in an ordered state, where the oily tails are neatly aligned, or a disordered state, where they are irregularly packed. Virtual 'heating' of the membrane caused it to shift from an ordered to a disordered state. When a simple transmembrane protein favoring the disordered state was inserted into the ordered state of the modeled membrane, disordered regions formed locally around the protein and the protein was able to diffuse within the membrane.

Modeling what would happen if two transmembrane proteins approached each other revealed that a consequence of the order–disorder transition is a strong attractive force that assembles the proteins together. Katira, Mandadapu, Vaikuntanathan et al. named this new phenomenon the 'orderphobic effect'. The forces arising from this effect were much greater than those currently believed to contribute to the assembly of membrane protein complexes, such as those generated by the elasticity of the membrane. This means that the orderphobic effect may be responsible for generating the protein clusters commonly seen in cell membranes.

Future work should next explore the opposite effect, where proteins favoring the ordered state are inserted into the disordered state of a membrane. This is expected to cause clustering of such proteins and thus large ordered regions in an otherwise disordered membrane.

favor disordered states. Importantly, the boundary of the domains resembles a stable, fluctuating order–disorder interface. The dynamic equilibrium established at the boundary allows the protein and its surrounding domain to diffuse. Moreover, because the interface has a finite stiffness, neighboring proteins can experience a membrane-induced force of adhesion, an attractive force that is distinctly stronger and can act over significantly larger lengths than those that can arise from simple elastic deformations of the membrane (*Dan et al., 1993*; *Goulian et al., 1993*; *Phillips et al., 2009*; *Kim et al., 1998*; *Haselwandter and Phillips, 2013*).

This force between transmembrane proteins is analogous to forces of interaction between hydrated hydrophobic objects. In particular, extended hydrophobic surfaces in water can nucleate vapor–liquid-like interfaces. In the presence of such interfaces, hydrophobic objects cluster to reduce the net interfacial free energy. This microscopic pre-transition effect manifesting the liquid–vapor phase transition can occur at ambient conditions (*Chandler, 2005*; *Lum et al., 1999*; *Willard and Chandler, 2008*; *Stillinger, 1973*; *ten Wolde and Chandler, 2002*; *Mittal and Hummer, 2008*; *Patel et al., 2011*; *2012*). In the transmembrane case, we show here that a protein favoring the disordered phase creates a similar pre-transition effect. In this case it manifests the order–disorder transition of a lipid bilayer. Like the raft hypothesis, therefore, clusters do indeed form, but the mechanism for their assembly and mobility emerge as consequences of order–disorder interfaces in an otherwise ordered phase. We refer to this phenomenon as the 'orderphobic effect'.

While considering the effect with one specific order–disorder transition, one should bear in mind its generic nature. The orderphobic effect should be a general consequence of a first-order transition, whether the transition is between solid-ordered and liquid-disordered phases as considered

explicitly herein, or between liquid-ordered and liquid-disordered phases as in multicomponent membrane systems. More is said on this point in the *Implications* section of this paper.

## The order–disorder transition is a first-order phase transition

We choose the MARTINI model of hydrated dipalmitoyl phosphatidylcholine (DPPC) lipid bilayers (*Marrink et al., 2007*) to illustrate the orderphobic effect. See *Materials and methods*. This membrane model exhibits an ordered phase and a disordered phase. *Figure 1A* contrasts configurations from the two phases, and it shows our estimated phase boundary between the two phases. The ordered phase has regular tail packing compared to the disorganized tail arrangement of the disordered phase. A consequence of the regular tail packing is that hydrophobic thickness of the ordered phase, $\mathcal{D}_o$ is larger than that of the disordered phase, $\mathcal{D}_d$. Correspondingly, the area per lipid in the ordered phase is smaller than that in the disordered phase.

Rendering the end particles of all the lipid chains in one of the two monolayers provides a convenient visual representation that distinguishes the two phases. These tail-end particles appear hexagonally-packed in the ordered phase and randomly arranged in the disordered phase. Regions that appear empty in this rendering are in fact typically filled by non tail-end particles or by tail-end particles from the other lipid monolayer.

To quantify the distinctions between the two phases, we consider a local rotational-invariant (*Nelson, 2002*; *Halperin and Nelson, 1978*; *Frenkel et al., 1980*), $\phi_l = |(1/6)\sum_{j\in\mathrm{nn}(l)} \exp(6\,i\,\theta_{lj})|^2$, where $\theta_{lj}$ is the angle between an arbitrary axis and a vector connecting tail-end particle $l$ to tail-end particle $j$, and the summation is over the six nearest neighbors of particle $l$. The equilibrium average, $\langle\phi_l\rangle$, is 1 for a perfect hexagonal packing, and it is 1/6 or smaller in the absence of bond-orientation correlations. Small periodically replicated samples of the hydrated DPPC membrane exhibit hysteretic changes in area per lipid and in $\langle\phi_l\rangle$ during heating and cooling. See *Appendix*, and *Marrink et al. (2005)* and *Rodgers et al. (2012)*. To establish whether the first-order-like behavior persists to large scales and thus actually manifests a phase transition, we consider larger systems and the behavior of the interface that separates the ordered and disordered phases.

*Figure 1B* shows coexistence for a system size of $N$ = 3900 lipids with an interface between the two phases. To analyze interfacial fluctuations, we first identify the location of the interface at each instant. This location is found with a two-dimensional version of the three-dimensional constructions described in *Limmer and Chandler (2014)* and *Willard and Chandler (2010)*. Specifically, and as discussed in *Materials and methods*, the interface is the line in the plane of the bilayer with an intermediate coarse-grained value of the orientational-order density,

$$\phi(\mathbf{r}) = \sum_l \phi_l \delta(\mathbf{r} - \mathbf{r}_l). \tag{1}$$

where $\mathbf{r}_l$ is the position of the $l$th tail-end particle projected onto a plane parallel to that of the bilayer, $\mathbf{r}$ is a two-dimensional vector specifying a position in that plane, and $\delta(\mathbf{r})$ is Dirac's delta function. We focus on this field rather than the tail-end number density, $\rho(\mathbf{r}) = \sum_l \delta(\mathbf{r} - \mathbf{r}_l)$, because the difference between the two phases is larger for typical orientational-order than for typical tail-end density.

A director density field, $u(\mathbf{r}) = \sum_l u_l \delta(\mathbf{r} - \mathbf{r}_l)$, could also be used to distinguish disordered regions from ordered regions. $u_l$ would specify the degree to which the hydrophobic chain of lipid $l$ is perpendicular to the average plane of the membrane. A field of this form would be useful for systems where liquid-ordered behavior occurs in the absence of solid-ordered behavior. Multicomponent membranes, for example, can exist in solid-ordered, liquid-ordered, and liquid-disordered states. For constructing the order–disorder interface of the simple one-component membrane considered here, however, $u(\mathbf{r})$ offers little more information than $\phi(\mathbf{r})$.

*Figure 1C* shows the Fourier spectrum of the height fluctuations of this interface, $\langle|\delta h_k|^2\rangle$. Two different system sizes are studied, with the larger system having approximately double the interface length of the smaller system. The Fourier component $\delta h_k$ is related to the height fluctuation $\delta h(x)$ as $\delta h(x) = \sum_k \delta h_k \exp(ikx)$ where $x$ is a point along the horizontal in *Figure 1B*. Here, $0 \leqslant x \leqslant L$, and $L$ is the box length. With periodic boundary conditions, $k = 2\pi m/L$, $m = 0, \pm 1, \pm 2, \cdots$. According to capillarity theory for crystal–liquid interfaces (*Nozières, 1992*; *Fisher et al., 1982*), $\langle|\delta h_k|^2\rangle \sim k_\mathrm{B}T/L\gamma k^2$ for small $k$, with $k_\mathrm{B}$ being Boltzmann's constant.

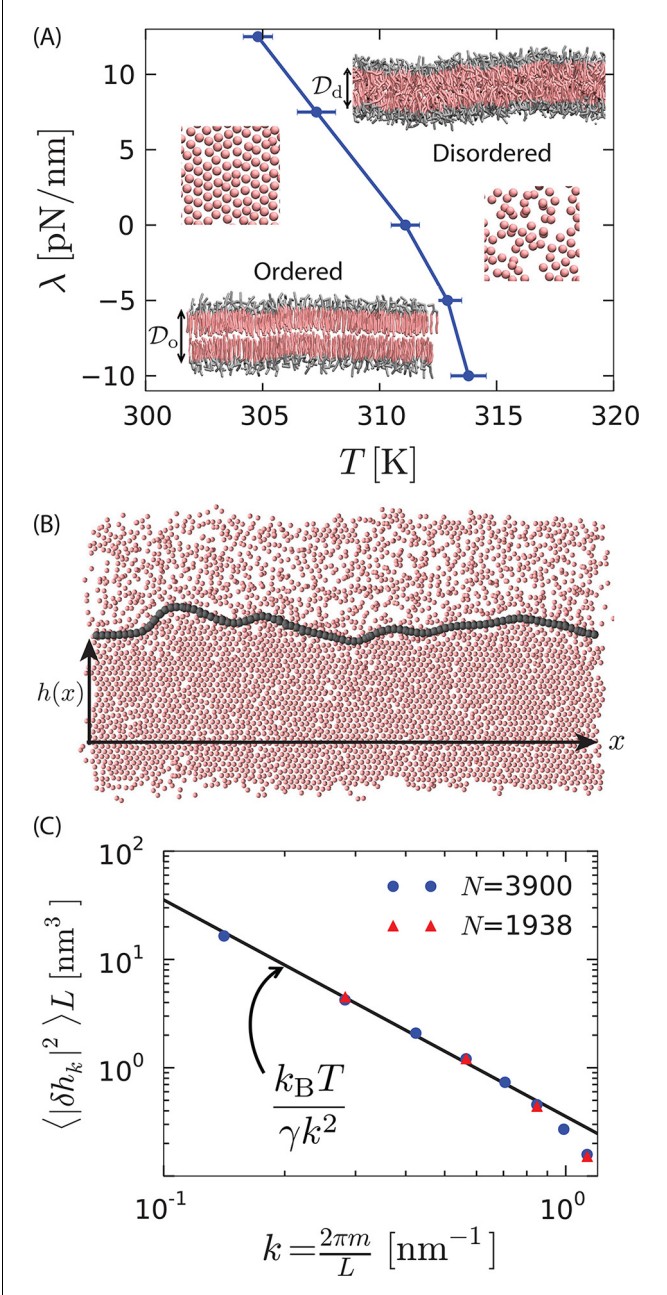

**Figure 1.** First-order phase transition in a model lipid bilayer. (**A**) Order–disorder phase diagram in the tension–temperature, $\lambda - T$, plane. The lateral pressure across the membrane is $-\lambda$. Points are estimated from 10 independent heating runs like those illustrated in *Appendix 1–figure 1* for a periodic system with 128 lipids. Insets are cross sections showing configurations of a bilayer with 3200 lipids in the ordered and disordered phases. The heads are colored gray while the tails are colored pink. Water particles are omitted for clarity. The hydrophobic thicknesses, $\mathcal{D}_{\mathrm{o}}$ and $\mathcal{D}_{\mathrm{d}}$, are the average vertical distances from the first tail particle of the upper monolayer to that of the lower monolayer. A macroscopic membrane buckles for all $\lambda < 0$. Snapshots of the last tail beads in one monolayer of each phase are shown to illustrate the difference in packing. (**B**) Snapshot of a system showing coexistence between the ordered and disordered phases. The gray contour line indicates the location of the interface separating the ordered and disordered regions. The snapshot is a top view of the bilayer showing the tail-end particles of each lipid in one monolayer. $h(x)$ is the distance of the instantaneous interface from a reference horizontal axis. (**C**) Fourier spectrum of $h(x)$. The line is the small-$k$ capillarity-theory behavior with $\gamma = 11.5$ pN.

Given the proportionality with $1/k^2$ at small $k$ (i.e., wavelengths larger than 10 nm), comparison of the proportionality constants from simulation and capillarity theory determines the interfacial stiffness (*Camley et al., 2010*), yielding $\gamma = 11.5 \pm 0.46$ pN. This value is significantly larger than the prior estimate of interfacial stiffness for this model, $3 \pm 2$ pN (*Marrink et al., 2005*). That prior estimate was obtained from simulations of coarsening of the ordered phase.

Because the ordered phase has a hexagonal packing, the interfacial stiffness depends on the angle between the interface and the lattice of the ordered phase. For a hexagonal lattice, there are three symmetric orientations for which the interfacial stiffnesses are equal. We will see that for the model we have simulated there appears to be only little angle dependence. Irrespective of that angle dependence, the stability of the interface and the quantitative consistency with capillary scaling provide our evidence for the order–disorder transition being a first-order transition in the model we have simulated.

The system sizes we have considered contain up to $10^7$ particles, allowing for membranes with $N \approx 10^4$ lipids, and requiring 10 μs to equilibrate. As such, our straightforward simulations are unable to determine whether the ordered phase is hexatic or crystal because correlation functions that would distinguish one from the other (*Nelson et al., 1982*) require equilibrating systems at least 10 times larger (*Bernard and Krauth, 2011*). Similarly, we are unable to determine the range of conditions for which the membranes organize with ripples and with tilted lipids (*Sirota et al., 1988*; *Smith et al., 1990*). Presumably, the ordered domain of the phase diagram in *Figure 1A* partitions into several subdomains coinciding with one or more of these possibilities. With advanced sampling techniques (*Frenkel and Smit, 2001*), free energy functions of characteristic order parameters can be computed to estimate the positions of boundaries between these various ordered behaviors. Here, we do not pursue this additional level of detail in the phase diagram because the additional boundaries refer to *continuous* transitions (*Sirota et al., 1988*). It is only the first-order transition, with its *discontinuous* change between ordered and disordered phases, that supports coexistence with a finite interfacial stiffness, and it is this stiffness that results in the orderphobic effect, which we turn to now.

## Transmembrane proteins can disfavor the ordered membrane

A disordering (i.e., orderphobic) transmembrane protein is one that solvates more favorably in the disordered phase than in the ordered phase. The disordering effect of the protein could be produced by specific side chain structures. See *Appendix*. Here, in the main text, we consider a simpler mechanism. In particular, we have chosen to focus on the size of the protein's hydrophobic thickness and the extent to which that thickness matches the thickness of the membrane's hydrophobic layer (*Killian, 1998*; *Sharpe et al., 2010*). See *Figure 2*.

The membrane's hydrophobic layer is thicker in the ordered state than in the disordered state. For instance, at zero lateral pressure and 294 K in the model DPPC membrane, we find that the average thicknesses of the hydrophobic layers in the ordered and disordered states are $\mathcal{D}_o = 3.1$ nm and $\mathcal{D}_d = 2.6$ nm, respectively. A transmembrane protein with hydrophobic thickness of size $\ell \approx 2.6$ nm will therefore favor the structure of the disordered phase. If the protein is large enough, it can melt the ordered phase near the protein and result in the formation of an order–disorder interface.

## Spatial variation of the order parameter field characterizes the spatial extent of the pre-melting layer

To evaluate whether a model protein is nucleating a disordered domain in its vicinity, we calculate the average of the orientational-order density field as a function of $r = |\mathbf{r}|$, $\langle \phi(r) \rangle$ (right axis of *Figure 2C*). It exhibits oscillations manifesting the atomistic granularity of the system. Dividing by the mean density $\langle \rho(r) \rangle$ largely removes these oscillations.

A profile of this ratio in the vicinity of the protein is depicted in *Figure 2C* (left axis). It changes approximately sigmoidally, connecting its values of 0.15 and 0.45 in the disordered and ordered phases, respectively. The shape of the profile suggests the formation of an order–disorder interface (*Rowlinson and Widom, 1982*). Further, the increase in the spatial extent of the disordered region with the increasing size of the protein, *Figure 2D*, is indicative of length scale dependent broadening effects brought about by capillary fluctuations. These impressions can be quantified by analyzing fluctuations of the instantaneous interface, which we turn to now.

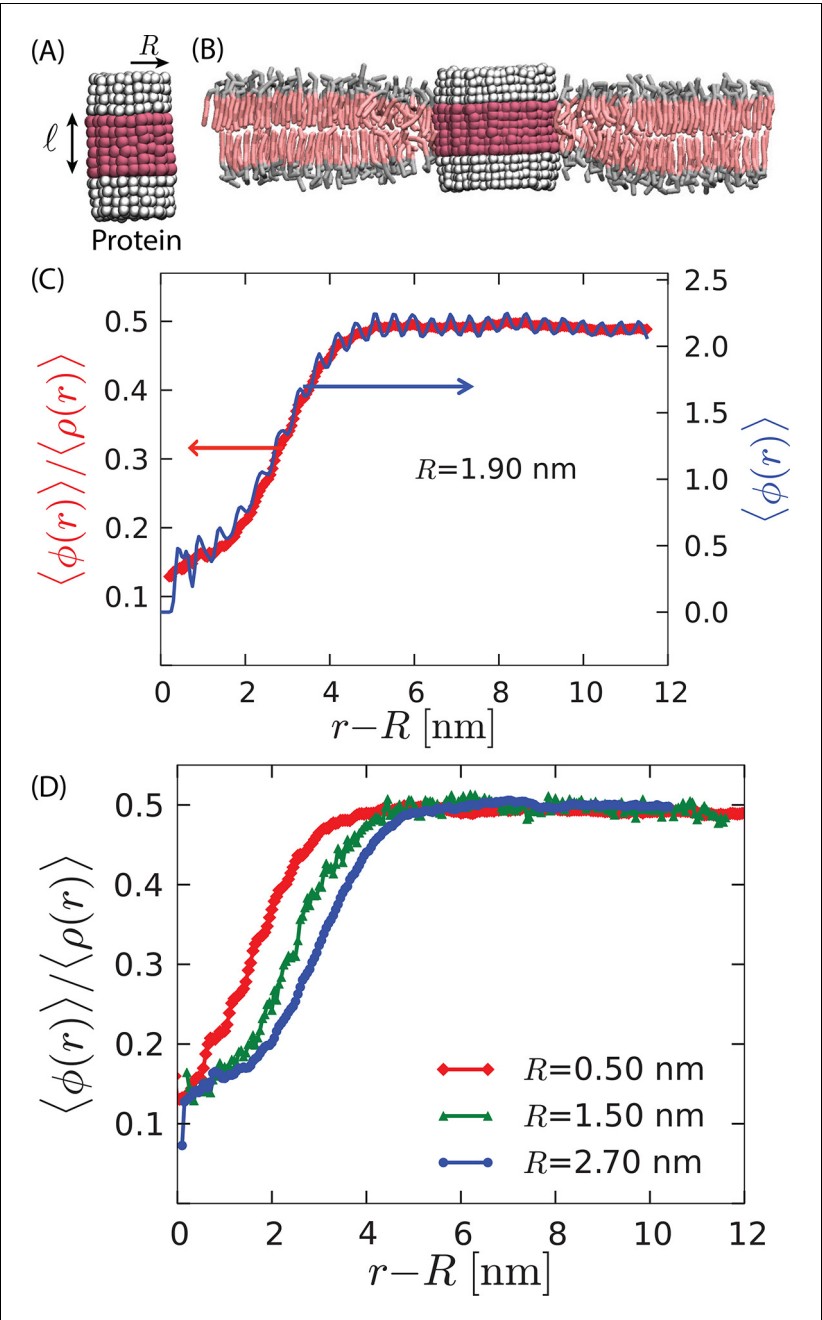

**Figure 2.** Model proteins in the bilayer. (**A**) Idealized cylindrical protein-like solutes with radius $R$ and hydrophobic thickness $\ell$ (magenta). The hydrophilic caps of the protein are shown in white. (**B**) Cross section of the lipid bilayer in the ordered phase containing a model protein of radius 2.7 nm with a hydrophobic thickness $\ell = 2.3$ nm $\leq \mathcal{D}_{\mathrm{d}}$. (**C**) The radial variation of the order parameters $\langle \phi(r) \rangle$ (right axis) and $\langle \phi(r) \rangle / \langle \rho(r) \rangle$ (left axis) show disorder in the vicinity of the protein of radius 1.9 nm. (**D**) Comparison of the radial order parameter variation for three different proteins shows an increase in the extent of the induced disorder region with protein radius.

## An orderphobic protein nucleates a fluctuating order–disorder interface

*Figure 3A* shows a configuration of the instantaneous interface that forms around the orderphobic protein shown in *Figure 2B*. The interface is identified as described above. A video of its dynamics is provided as *Video 1*. As is common in crystal–liquid interfaces, the interface nucleated by an

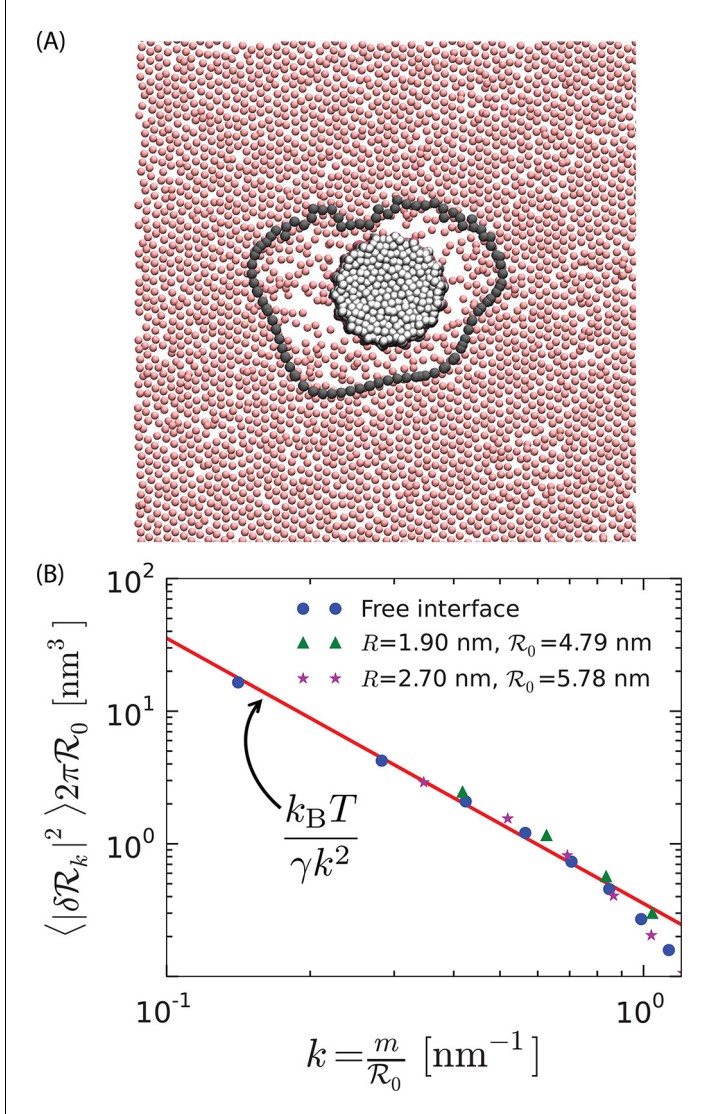

**Figure 3.** Soft order–disorder interface. (**A**) Arrangement of the tail-end particles of the top monolayer corresponding to the protein in *Figure 2B*. Far away from the protein, the tail-end particles show hexagonal-like packing and are in the ordered state. Proximal to the protein, it can be seen that the tail-end particles are randomly arranged, and resemble the disordered phase. The line connected by the black points denotes the instantaneous order–disorder interface. (**B**) The fluctuations in the radius of the order–disorder interface are consistent with the fluctuations of a free order–disorder interface at coexistence. $\mathcal{R}_0$ is the mean radius of the order–disorder interface surrounding a model protein of radius $R$.

orderphobic protein may exhibit hexagonal faceting (*Nozières, 1992*), remnants of which can be observed in *Figure 3A*.

The mean interface is a circle of radius $\mathcal{R}_0$. Fourier analysis of fluctuations about that circle yields a spectrum of components. To the extent that these fluctuations obey statistics of capillary wave theory for a circular interface, the mean-square fluctuation for the $k$th component is $\langle|\delta\mathcal{R}_k|^2\rangle = k_{\mathrm{B}}T/2\pi\gamma k^2\mathcal{R}_0$, where $k = m/\mathcal{R}_0$ and $m = \pm1, \pm2, \cdots$, and $\gamma$ is the order–disorder interfacial stiffness, neglecting the dependence on the angle between the interface and the lattice. The discrete values of $k$ reflect periodic boundary conditions going full circle around the model protein.

In *Figure 3B*, we use the interfacial stiffness from the free interface ($\gamma = 11.5$ pN) separating coexisting ordered and disordered phases with the capillary theory expression, and its corresponding spectrum, to compare with the spectrum of the protein-induced interface. The agreement between

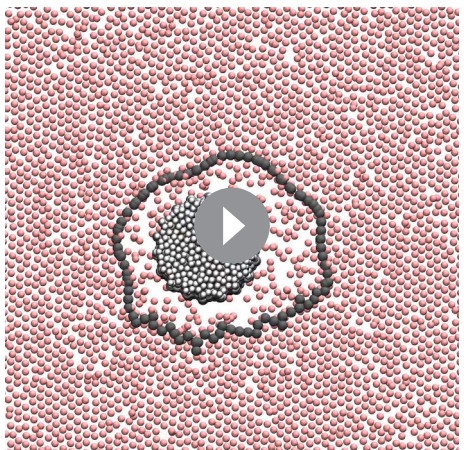

**Video 1.** Instantaneous interface around an orderphobic protein. Also uploaded to https://goo.gl/NBQJP9.

the theory, the free interface and the protein-induced interface is good, and it improves as the radius of the orderphobic protein increases and the wave vector $k$ decreases. This agreement indicates that the orderphobic protein does indeed nucleate an interface manifesting the order–disorder transition. The deviations of the fluctuations of the free interface from capillary wave theory occur for $k \gtrsim 0.8$ nm$^{-1}$, corresponding to wavelengths $2\pi/k \lesssim 7$ nm, and a mean interface radius $\mathcal{R}_0 \lesssim 1$ nm. Indeed, **Figure 2** suggests that even a small protein of radius 0.5 nm, which supports an interface of radius $\mathcal{R}_0 \approx 1.2$ nm, is sufficient to induce an order–disorder interface with fluctuations consistent with capillary theory.

## The orderphobic effect generates forces of assembly and facilitates protein mobility

*Figure 4* shows three snapshots from a typical trajectory initiated with two orderphobic proteins of radius 1.5 nm separated by a distance of 14 nm. Each induces a disordered region in its vicinity, with soft interfaces separating the ordered and disordered regions. The free energy of the separated state is approximately $\gamma(P_1 + P_2)$, where $P_i$ is the perimeter of the order–disorder interface around protein $i$. On average, $\langle P_i \rangle = 2\pi\mathcal{R}_0$. After a few hundred nanoseconds, a fluctuation occurs where the two interfaces combine. While the single large interface remains intact, the finite tension of the interface pulls the two proteins together. Eventually, the tension pulls the two proteins together with a final perimeter, $P_f$, that is typically much smaller than $P_1 + P_2$. A video of its dynamics is provided as *Video 2*.

After the separated interfaces join, the assembly process occurs on the time scale of microseconds. This time is required for the proteins to push away lipids that lie in the path of the assembling proteins. Given this time scale, a reversible work calculation of the binding free energy would best control both the distance and the number of lipids between the proteins. Moreover, the evident role of interfacial fluctuations indicates that the transition state ensemble for assembly must involve an interplay between inter-protein separations and lipid ordering as well as lipid concentration.

While we leave the study of reversible work surfaces and transition state ensembles to future work, it seems already clear that the net driving force for assembly is large compared to thermal

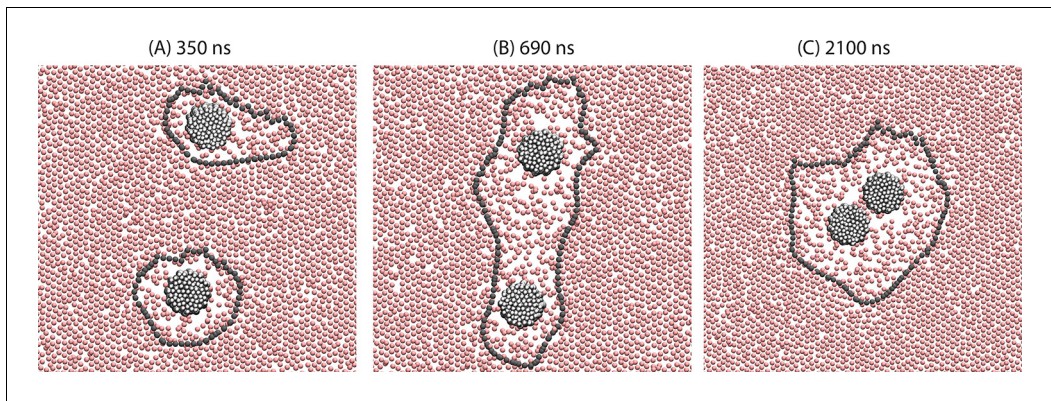

**Figure 4.** Demonstration of the orderphobic force: two proteins separated by a center-to-center distance of 14 nm are simulated at 309 K. Snapshots at various times reveal the process of assembly in which the two order–disorder interfaces merge into a single interface.

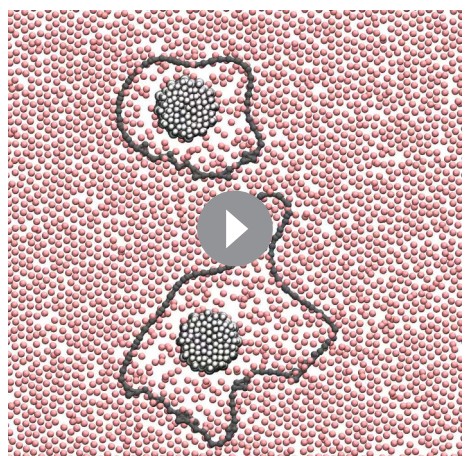

**Video 2.** Assembly of two orderphobic proteins. Also uploaded to https://goo.gl/HXS0j7.

energies. For example, with a model orderphobic protein radius of 1.5 nm, we find $\gamma(\langle P_{\mathrm{f}} \rangle - 2\langle P_1 \rangle) \approx -30\,k_{\mathrm{B}}T$. The range over which the force acts is given by the average radius of the two interfaces, $2\mathcal{R}_0$. This range is further amplified by the width of the interface, which is of $\mathcal{O}(\mathcal{R}_0)$ for one-dimensional interfaces in two-dimensional systems (*Kardar, 2007*). The typical range is $\approx 10$ to 30 nm. In comparison, given the elastic moduli of the membranes we consider, elastic responses will generate attractive forces between transmembrane proteins that are much smaller in strength and range, typically $-5\,k_{\mathrm{B}}T$ and 1 nm, respectively (*Haselwandter and Phillips, 2013*; *de Meyer et al., 2008*). Moreover, similarly weak and short ranged forces are found from solvation theory that accounts for linear response in microscopic detail while not accounting for the possibility of an underlying phase transition (*Lague et al., 1998*).

As in the hydrophobic effect (*Chandler, 2005*), the strength and range of the orderphobic force leverages the power of a phase transition, depending in this case on the ability of the orderphobic protein to induce a disordered layer in its vicinity. This ability depends upon the proximity to the membrane's phase transition, and, for the simple protein models considered in this paper, it depends upon the protein's radius and hydrophobic mismatch with the membrane. The spatial extent of the disordered region increases with proximity to phase coexistence as shown in *Figure 5A*.

Furthermore, *Figure 5B* shows that the strength of the effect is maximal for a hydrophobic thickness equal to that of the disordered phase, and it decreases as the hydrophobic thickness approaches that of the ordered phase. In the case of zero mismatch (i.e., $\ell = \mathcal{D}_{\mathrm{o}}$) the value of the order parameter in the vicinity of the protein is consistent with that of a pure bilayer in the ordered state. Therefore, the model proteins with zero mismatch do not induce a disordered region, and the orderphobic effect vanishes. See *Figure 5B and D*.

*Figure 4* also shows that the orderphobic effect produces excess mobility, by proteins melting order in a surrounding microscopic layer and by facilitating the motions of neighboring proteins. This finding explains how protein mobility and reorganization can be relatively facile in the so-called 'gel' phases of membranes. Further information on this phenomenon is provided in *Appendix*. Our prediction of enhanced lipid mobility surrounding orderphobic proteins may be amenable to experimental tests by single molecule tracking techniques (*Eggeling et al., 2009*).

## Implications of the orderphobic effect and related phenomena in biological membranes

Biological membranes and transmembrane proteins are far more complicated than the models considered in this paper. Part of the complexity is associated with multiple components, which allow for more than one order–disorder transition. For example, with a membrane composed of three components, coexistence can be established between liquid-ordered and liquid-disordered phases (*Veatch and Keller, 2005*), and both of these phases exist in bio-membranes (*Swamy et al., 2006*; *Owen et al., 2012*; *Polozov et al., 2008*). The fact that liquid-ordered and liquid-disordered phases can coexist with finite line tension (*Veatch and Keller, 2005*) implies the existence of a first-order transition between them (*Chandler, 1987*) and thus the relevance of the orderphobic effect. This effect is much wider in applicability than the Casimir effect (*Machta et al., 2012*), which applies only within the much smaller range of conditions where the first-order transition reaches its limiting case of criticality. A director density for hydrophobic chains serves as the order parameter distinguishing liquid-ordered and liquid-disordered phases. The strength and range of orderphobic effects that will arise from this order–disorder transition merit future investigation. Modeling might build from recent numerical work on the liquid-ordered phase (*Risselada and Marrink, 2008*).

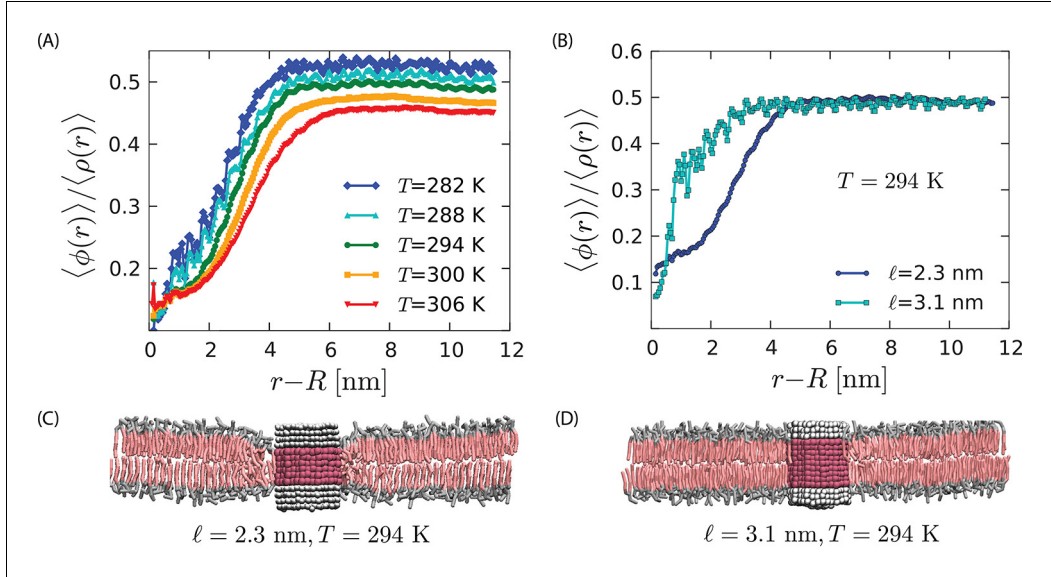

**Figure 5.** Strength of the orderphobic force. (**A**) Radial variation of the order parameter showing the extent of the disordered region as a function of temperature, for a protein of radius 1.9 nm and hydrophobic thickness 2.3 nm. The extent of the disordered region increases as the melting temperature is approached, at zero surface tension. (**B**) Comparison of the radial variation of the order parameter for different hydrophobic mismatches. Proteins with no mismatch do not create any disordered region. (**C**) Arrangement of lipids around a protein with negative mismatch. (**D**) Arrangement of lipids around a protein with zero mismatch.

Bear in mind that the strength and range of the orderphobic effect depends upon the proximity of the order–disorder transition. This proximity can be changed by changing temperature, as illustrated in *Figure 5A*. With many components in play, the proximity can also be changed by varying membrane composition. One can therefore anticipate that the strength and range of orderphobic effects will depend upon, for example, cholesterol concentrations. It will also depend upon the presence of additional proteins, and the domains formed with those proteins themselves depend upon the orderphobic effect.

Another source of complexity is the side-chain structure of transmembrane proteins. These side chains can affect the packing of lipid chains. To the extent that lipid packing is disrupted, even small $\alpha$-helix proteins can be orderphobic. Evidence for this assessment is provided in *Appendix*. Thus, the orderphobic effect can lead to clustering of transmembrane $\alpha$-helices. Moreover, just as the strength and range of the orderphobic effect can be modified by changing the radius and mismatch of our model proteins, the strength and range of the orderphobic effect will also be affected by the structure of protein side chains. Further, an obvious consequence of the orderphobic effect is the existence of a driving force that will move orderphobic proteins from an ordered phase to a disordered phase, and the creation of large disordered domains as a result of clustering orderphobic proteins. Both of these effects have been noted in simulations of disordering $\alpha$-helix proteins in a membrane exhibiting coexisting liquid-ordered and liquid disordered domains (*Schäfer et al., 2011*; *Domański et al., 1818*).

Further, there is a dual to the orderphobic effect: a transmembrane protein in the disordered phase that favors the ordered phase can nucleate an ordered region and order–disorder interface. For example, one of our model proteins with a positive mismatch ($\ell = \mathcal{D}_{o}$) would induce order in its vicinity. This effect is illustrated in the *Appendix*. Interfaces separating the ordered and disordered regions will again provide a force for assembly. This case corresponds to the situation of lipid rafts (*Simons and Ikonen, 1997*), which consists of ordered domains floating in otherwise disordered membranes. The stable interface separating domains then serves as a concrete geometrical definition of the raft. This orderphilic effect will depend upon the extent to which the surface of the transmembrane protein is commensurate with the ordered phase structure. Hydrophobic mismatch is but one possibility. $\beta$-sheets that align neighboring lipids are others. The fact that the orderphilic effect

is a pre-transition effect for the first-order transition between ordered and disordered phases implies it should occur in disordered membranes that are thermodynamically close to coexistence between liquid-ordered and liquid-disordered phases.

The orderphobic effect may also be of direct relevance in understanding the behavior of lung-surfactant monolayers. The primary component of these monolayers is the lipid DPPC, with melting temperature higher than physiological temperature (41°C), and a small proportion of cholesterol, and proteins. These monolayers undergo cyclic surface tension mediated phase transitions between the ordered and disordered phases (*Nag et al., 1998*). The results of this paper are also applicable to lipid monolayers and could govern the diffusion and assembly of proteins embedded within the relatively rigid ordered phases.

Finally, we speculate that the orderphobic effect plays important roles in membrane fusion, budding, and cell signaling (*Fratti et al., 2004*; *Zick et al., 2014*; *Qi et al., 2001*; *James and Vale, 2012*; *Różycki et al., 2012*). In the case of fusion, it would appear that one important role is to promote fluctuations in an otherwise stable membrane. Otherwise, it is difficult to conceive of a mechanism by which thermal agitation would be sufficient to destabilize microscopic sections of membranes. Such destabilization seems necessary for initiating and facilitating membrane fusion. Many proteins are involved in such processes (*Fratti et al., 2004*; *Fasshauer et al., 1998*; *Wickner and Schekman, 2008*), but it may not be a coincidence that the hydrophobic thicknesses of SNARE proteins are 25% smaller than that of the ordered membrane states (*Milovanovic et al., 2015*; *Stein et al., 2015*).

## Materials and methods

### Molecular simulations

We simulate the MARTINI coarse-grained force field using the GROMACS molecular dynamics package (*Marrink et al., 2007*; *Pronk et al., 2013*). 'Antifreeze' particles are added to the solvent to ensure that the solvent does not freeze over the temperature range considered in the simulations as in *Marrink et al. (2007)*. Thermostats and barostats control temperature and pressure, and checks were performed to assure that different thermostats and barostats yielded similar results (*Frenkel and Smit, 2001*). The hydrophobic cores of our idealized proteins are constructed using the same coarse-grained beads as the lipid tails (particle C1 in the MARTINI topology *Marrink et al. (2007)*). Similarly, the hydrophilic caps are constructed using the first bead of the DPPC head group (Q0, in the MARTINI topology). The protein beads also have bonded interactions where the bond length is 0.45 nm and the bond angle is set to 180°. The associated harmonic force constants for the bond lengths and angles are 1250 kJmol$^{-1}$nm$^{-2}$ and 25 kJmol$^{-1}$rad$^{-2}$. Based on the hydrophobic mismatch with the bilayers, the proteins are classified into three categories: (i) positive mismatch ($\ell > \mathcal{D}_{\mathrm{o}}$) (ii) negative mismatch ($\ell \leq \mathcal{D}_{\mathrm{d}}$) and (iii) no mismatch ($\ell \approx \mathcal{D}_{\mathrm{o}}$). To create different mismatches, we alter the number of beads in the protein core. These idealized proteins do not contain charges.

Proteins are embedded in the equilibrated bilayer at 279 K. The resulting system is then heated to the required temperature and equilibrated for another 1.2 µs. All the subsequent averages are performed using 10 independent trajectories each 600 ns long. The assembly of proteins is also performed using the same DPPC bilayer system with 3200 lipids and 50000 water beads. In this case, two proteins are inserted in this bilayer with centers at a distance of 14 nm and the simulation is carried out at 309 K.

The flat interface is stabilized by juxtaposing an ordered bilayer equilibrated at 285 K and zero lateral pressure with a disordered bilayer equilibrated at the same conditions corresponding to the cooling and heating curves of the hysteresis loop in *Appendix 1—figure 1*, respectively. The system thus constructed is equilibrated in the ensemble with fixed temperature, volume, and numbers of particles. This ensemble allows for maintaining an area per lipid intermediate between the two phases, thus stabilizing the interface.

### Instantaneous interface

For the purpose of obtaining a smooth and continuous interface, $\phi(\mathbf{r})$ is coarse grained by replacing Dirac's delta function with a finite-width Gaussian, $(1/2\pi\xi^2) \exp\left(-|\mathbf{r}|^2/2\xi^2\right)$. The replacement

changes $\phi(\mathbf{r})$ to $\bar{\phi}(\mathbf{r})$. The coarse-graining width, $\xi$, is chosen to be the average separation between tail-end particles $l$ and $j$ when $\langle(\phi_l - \langle\phi_l\rangle)(\phi_j - \langle\phi_j\rangle)\rangle/\langle(\phi_l - \langle\phi_l\rangle)^2\rangle$ in the ordered phase is 1/10. This choice yields a value of $\xi = 1.5$ nm. The instantaneous order–disorder interface is the set of points s satisfying $\bar{\phi}(\mathbf{s},t) = (\phi_{\mathrm{d}} + \phi_{\mathrm{o}})/2$. Here, $\phi_{\mathrm{d}}$ and $\phi_{\mathrm{o}}$ are $\langle\phi(\mathbf{r})\rangle$ evaluated in the disordered and ordered phases, respectively. At zero lateral pressure and 294 K, we find $\phi_{\mathrm{d}} = 0.4 \pm 0.02$ nm$^{-2}$ and $\phi_{\mathrm{o}} = 2.15 \pm 0.2$ nm$^{-2}$. For numerics, a square lattice tiles the average plane of the bilayer, and the coarse-grained field $\bar{\phi}(\mathbf{r})$ is evaluated at each lattice node. Values between are determined by interpolation. For convenience, the Gaussian function is truncated and shifted to zero at $3\xi$. Any value of $\xi$ within the range, 1 nm $< \xi <$ 2 nm gives nearly identical $\bar{\phi}(\mathbf{r})$. Outside that range, larger values obscure detail by excessive smoothing, and smaller values obscure detail by capturing a high density of short-lived bubbles of disorder.

## Acknowledgements

It is a pleasure to thank Axel T Brunger, Jay Groves, John Kuriyan, and Sarah Keller for helpful conversations about this work, and Daan Frenkel, Tom Lubensky, and Siewert-Jan Marrink for helpful comments on earlier drafts of this work.

## Additional information

### Funding

| Funder | Grant reference number | Author |
| --- | --- | --- |
| U.S. Department of Energy | DE-AC02-05CH11231 | Kranthi K Mandadapu<br>Suriyanarayanan Vaikuntanathan<br>David Chandler |
| Lawrence Berkeley National Laboratory | | Shachi Katira<br>Kranthi K Mandadapu<br>Suriyanarayanan Vaikuntanathan<br>Berend Smit<br>David Chandler |
| University of Chicago | | Suriyanarayanan Vaikuntanathan |
| U.S. Department of Energy | FWP number SISGRKN | Shachi Katira<br>Berend Smit |

SV is currently supported by the University of Chicago. In the initial stages, he was supported by Director, Office of Science, Office of Basic Energy Sciences, Materials Sciences, and Engineering Division, of the U.S. Department of Energy under contract No.\ DE AC02-05CH11231. KKM, DC, SK and BS are supported by that same DOE funding source, the latter two with FWP number SISGRKN. This research used resources of the National Energy Research Scientific Computing Center, a DOE Office of Science User Facility supported by the Office of Science of the U.S. Department of Energy under Contract No. DE-AC02-05CH11231 and resources of the Midway-RCC computing cluster at University of Chicago. The funders had no role in study design, data collection and interpretation, or the decision to submit the work for publication.

### Author contributions

SK, Conception and design, Acquisition of data, Analysis and interpretation of data, Drafting or revising the article, Contributed unpublished essential data or reagents; KM, Conception and design, Acquisition of data, Analysis and interpretation of data, Drafting or revising the article, Contributed unpublished essential data or reagents; SV, Conception and design, Acquisition of data, Analysis and interpretation of data, Drafting or revising the article, Contributed unpublished essential data or reagents; BS, Conception and design, Acquisition of data, Analysis and interpretation of data, Drafting or revising the article, Contributed unpublished essential data or reagents; DC, Conception and design, Acquisition of data, Analysis and interpretation of data, Drafting or revising the article, Contributed unpublished essential data or reagents

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

## Appendix

In this Appendix, we examine hysteresis in the bilayer system, show results for mean-square displacements as functions of time, and provide evidence that a small transmembrane $\alpha$-helix protein can be orderphobic if the side chains of the protein disrupt the packing of lipid tails.

## The lipid bilayer system exhibits hysteresis

*Appendix 1—figure 1* shows the change in area per lipid with temperature while heating and cooling a bilayer. There are finite jumps in area per lipid as the system transitions between the two phases, suggesting a first-order phase transition. Hysteresis occurs because ordering from the metastable disordered phase is much slower than disordering from the metastable ordered phase. Due to the difference in time scales, when contrasting melting and freezing from heating and cooling runs, the melting points from heating runs as shown in *Appendix 1—figure 1A* provide the more accurate estimates of the actual phase boundaries. Systematic errors due to small system size and heating rate have not been estimated.

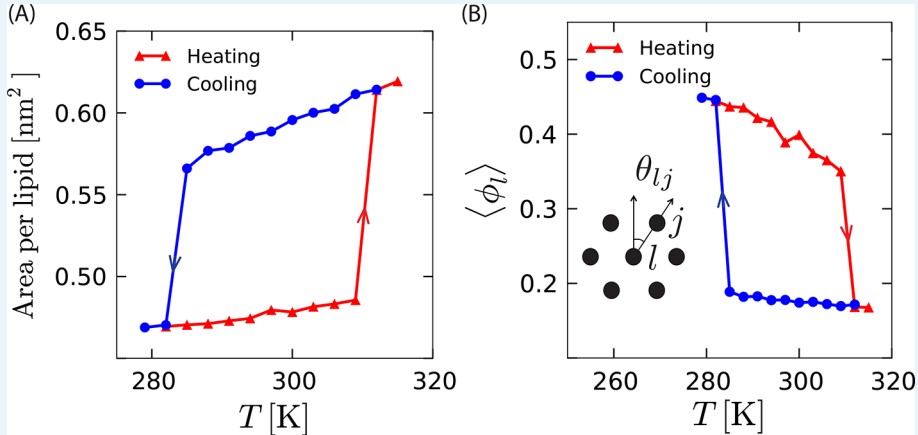

**Appendix 1—figure 1.** Structural measures of different phases as a function of temperature, $T$. (**A**) Variation in area per lipid with temperature during heating and cooling shows finite jumps and hysteresis. (**B**) Average local orientational order, $\langle \phi_l \rangle$, also shows finite jumps as a function of temperature while heating and cooling. Magnitudes of heating and cooling rates are 3 K/μs.

## The pre-melting layer has a higher mobility than the ordered phase

In *Appendix 1—figure 2* we show the mean squared displacements of lipids in the bulk ordered phase, bulk disordered phase, and the pre-melting layer induced by an orderphobic protein. As discussed in the main text, an orderphobic protein increases the mobility of the lipids in its vicinity. Note that the center of mass of the membrane fluctuates in time. These fluctuations affect the absolute positions of lipid molecules, but they are irrelevant to the issue of lipid mobility. Therefore, the mean-square displacements considered in *Appendix 1—figure 2* are for tail-end particle positions relative to the instantaneous

position of the membrane's center of mass. That is to say, for $\langle |\bar{\mathbf{r}}_l(t) - \bar{\mathbf{r}}_l(0)|^2 \rangle$, where $\bar{\mathbf{r}}_l(t)$ is the position at time $t$ of the $l$th tail-end particle less that of the membrane's center of mass.

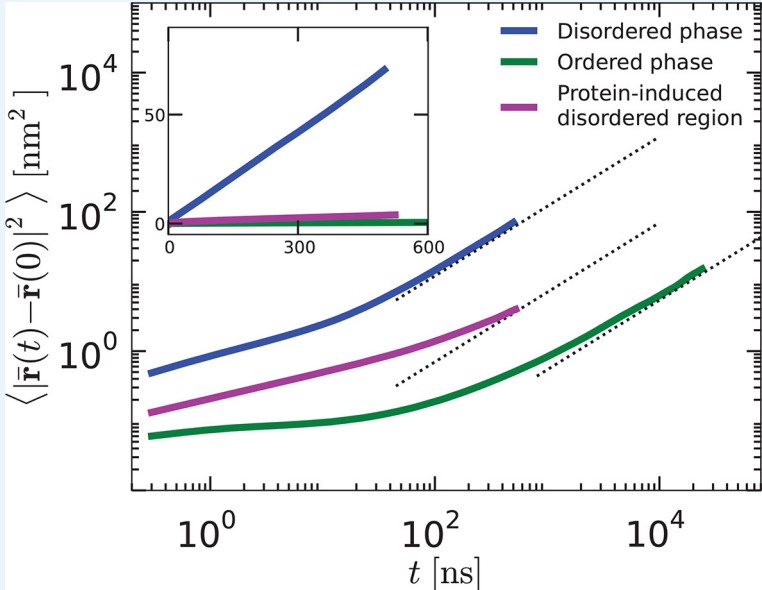

**Appendix 1—figure 2.** Mean-square displacements as functions of time, $t$, for lipids in the disordered phase, the protein-induced disordered domain, and the ordered phase. The functions are shown on log–log scale (main graphs) and linear scales (inset).

For large enough times, $t$, the mean-square displacements are asymptotic to $4Dt$, where $D$ is the self-diffusion constant. For the disordered liquid phase, we see from **Appendix 1—figure 2** that the asymptotic region is reached within $10^2$ ns, and that $D \approx 4 \times 10^{-7}$ cm$^2$/s. In contrast, for the ordered phase, the diffusive asymptotic limit is not reached until $10^4$ ns, and $D \approx 2 \times 10^{-9}$ cm$^2$/s. The mobility of lipids within the disordered layer surrounding the orderphobic protein is an order of magnitude larger than that of lipids beyond that region and in the ordered phase.

Experimental results for lipid diffusion constants in disordered and ordered bilayers are $D \approx 3 \times 10^{-8}$ cm$^2$/s and $D \approx 2 \times 10^{-10}$ cm$^2$/s, respectively (**Korlach et al., 1999**). The simulation is in harmony with experiment for the two-order of magnitude difference between the ordered- and disordered-phase values of $D$. Of course, absolute values of $D$ are beyond the scope of what can be predicted from our simulations because coarse graining omits degrees of freedom that would increase friction and decrease $D$.

## A model $\alpha$-helix is orderphobic

Here, we consider the MARTINI model for KALP23—a polypeptide chain with 23 residues, consisting of alanines and leucines flanked by lysines (**Schäfer et al., 2011**). This molecule has a hydrophobic thickness of $\ell \approx 3.0$ nm, which means that it has essentially no hydrophobic-length mismatch with the ordered bilayer. Nevertheless, it is orderphobic because its side chains perturb the ordered lipid phase to an extent that a pre-melting layer is formed around the protein. This behavior is demonstrated with the aid of **Appendix 1— figure 3**. The panels render configurations from a simulation in which we have placed this model protein in the MARTINI model for the ordered DPPC bilayer system considered in the main text. The pre-melting layer that forms around the protein causes the protein to tilt so as to keep its full hydrophobic length in contact with hydrophobic tails of the lipids. The

interface separating its disordered domain from the surrounding ordered phase remains
stable throughout a molecular dynamics trajectory running for more than 1 μs.

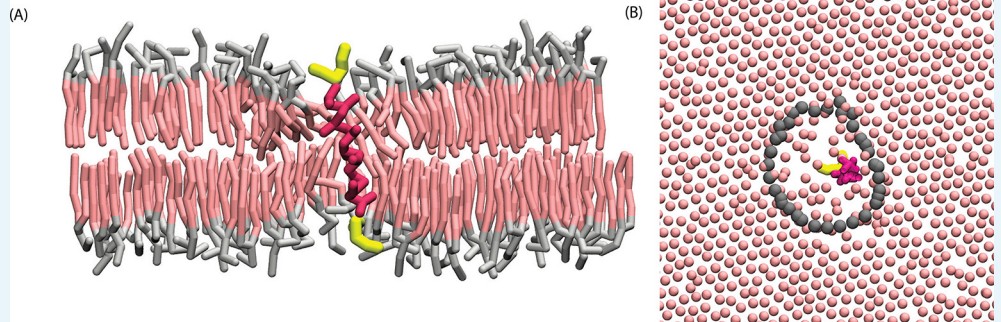

**Appendix 1—figure 3.** A model $\alpha$-helix is orderphobic. (**A**) Cross section of ordered phase of a
hydrated DPPC membrane containing one transmembrane KALP23 protein. Solvent water is
not rendered for purpose of clarity. Configuration was obtained after running simulation for
roughly 1 μs at 294 K and zero lateral pressure. (**B**) Configuration of tail-end particles for the
top monolayer, with gray points locating the instantaneous interface.

## An orderphilic protein in the disordered phase nucleates an ordered domain

In a disordered bilayer, we embed a model protein with a hydrophobic thickness approximately
equal to that of the ordered phase, $\ell = 3.3$ nm. This embedding illustrates the orderphilic
case. As a control, we also embed a model protein with a hydrophobic thickness
approximately equal to that of the disordered phase, $\ell = 2.3$ nm. For both cases, we then
calculate the average of the bond orientational order density $\langle \phi(r) \rangle$, and the number density
$\langle \rho(r) \rangle$. We also compute the director density, $\langle u(r) \rangle$ as defined earlier, with
$u_l = (3/2)\cos^2(\theta_l) - (1/2)$. Here, $\theta_l$ is the angle between the membrane normal and the $l$th
chain's orientation. The director density for $\ell = 3.3$ nm as seen in *Appendix 1—figure 4*
changes approximately sigmoidally, connecting its values of 0.92 and 0.62 in the ordered
and disordered phases, respectively. Further, we have computed the instantaneous
interfaces from the coarse graining of each of these three fields $\rho(\mathbf{r})$, $\phi(\mathbf{r})$ and $u(\mathbf{r})$. In the
orderphilic case, interfaces are apparent for all three fields. The director field provides the
clearest pictures, which are illustrated in *Appendix 1—figures 4,5*. In the control case,
interfaces do not appear.

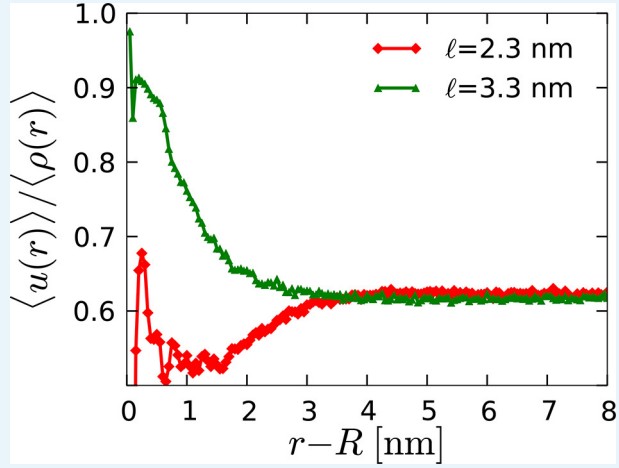

**Appendix 1—figure 4.** Radial profile of the average director density surrounding orderphobic

($\ell = 2.3$ nm) and orderphilic ($\ell = 3.3$ nm) model proteins in the disordered membrane phase.

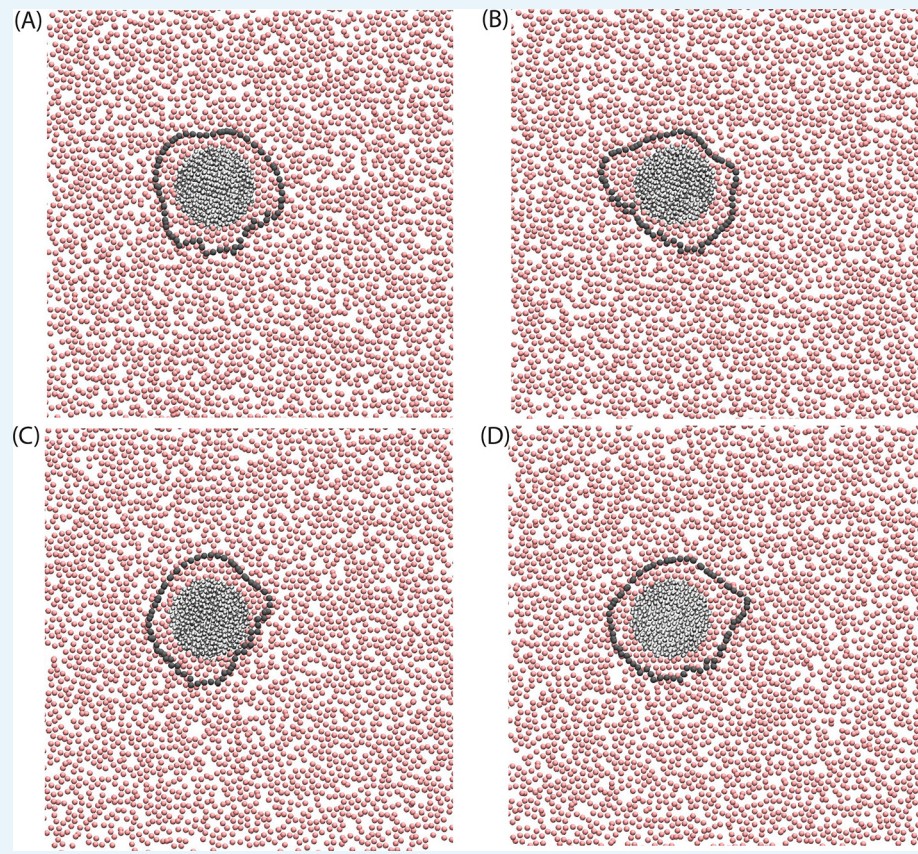

**Appendix 1—figure 5.** Configurations of the disordered membrane in the presence of a model orderphilic protein, $\ell = 3.3$ nm. The rendered particles are the 'C2' tail particles of the lipids, and the gray line marks the boundary between ordered and disordered domains by rendering the contour of the instantaneous interface for the director field, $u(\mathbf{r})$.

Distinctions between interfaces for each of the fields are microscopic effects worthy of future study. For the bulk interface between the ordered and disordered phases, the distinction disappears because there is only one order–disorder transition for the membrane model we have considered. In more complicated membrane models, those with mixtures of components exhibiting both liquid-ordered and solid-ordered phases, the different fields offer different information for both large and small length scales.

