## [Decision Letter]

Thank you for submitting your work entitled "Pre-transition effects mediate forces of assembly between transmembrane proteins" for consideration by *eLife*. Your article has been reviewed by 3 peer reviewers, one of whom is a member of our Board of Reviewing Editors, and the evaluation has been overseen by a Reviewing Editor and Richard Aldrich as the Senior Editor. Two of the three reviewers, Gerhard Hummer and Siewert-Jan Marrink, have agreed to share their identity. The reviewers have discussed the reviews with one another and the Reviewing Editor has drafted this decision to help you prepare a revised submission.

This work introduces a new concept to understand membrane protein associations. It suggests that clustering of membrane proteins may not be a consequence of inter-protein or protein-lipid "mismatches" (e.g., membrane thickness as suggested by Milovanovic et al. Hydrophobic mismatch sorts SNARE proteins into distinct membrane domains. Nat Commun 6:5984. doi: 10.1038/ncomms6984, 2015), but rather by the differences in membrane thickness in difference "phases" of a membrane. If transmembrane domains are shorter than the thickness of a particular phase, they would then prefer to be in a phase that matches the length of the transmembrane domains, hence clustering in these areas. Thus, clustering of certain membrane proteins (such as SNAREs) may be a consequence of this "order-phobic" (but see comment 6 below) effect.

More specifically, in this work, simulations of coarse-grained lipid membrane models and of membrane/protein models show that (1) transmembrane proteins can induce a local order/disorder transition in a lipid membrane at conditions close to phase coexistence, and (2) that such local phase separation can drive protein association by a reduction of the interfacial free energy between the ordered and disordered lipid phases. These effects should be general in lipid membranes at conditions close to an order/disorder transition. By using orientational order parameters, the phase boundary in the membrane could be clearly resolved, enabling a characterization of its properties. In particular, the fluctuations of the length and shape of the boundary are shown to follow a capillary wave model down to almost molecular length scales. The observation of nanoscale phase separation induced by transmembrane proteins perturbing the local structure should be of interest not only to the physico-chemical community but also to bioscientists. In particular, it may play a role in lipid-mediated assembly of integral membrane protein complexes and supercomplexes.

While the reviewers and editors found this work interesting, concerns were raised as outlined below that need to be addressed in a revised version before a decision can be made. In particular, the authors are asked to respond to the concern about the biological relevance of the particular phase transition. If possible the authors are encouraged to perform a simulation that is more relevant to biology.

1) The authors mention in the *Introduction* a number of papers (Nishimura et al., 2006; Polozov et al., 2008; Swamy et al., 2006; Munro, 2003; Thewalt and Bloom, 1992; Owen et al., 2012) that supposedly provide evidence for the physiological relevance of gel domains. Either the referenced papers deal with situations in which membranes are being cholesterol depleted or thermally quenched, or they talk about liquid-ordered domains, not gel domains. In fact, in Munro, 2003 it literally states "The solid gel phase is not thought to be of physiological relevance". Please change the Introduction accordingly.

2) The demonstration of the attraction between proteins in this manuscript concentrates on proteins that induce local disorder in an otherwise ordered phase. In biological membranes the disordered phase is expected to dominate, to ensure fluidity and facile transport. The authors may thus want to discuss the reversed situation in a bit more detail (which should be quite symmetric). Ideally, the authors are encouraged to consider a simulation that is more relevant for the biological situation.

3) The authors may want to discuss in more depth the relation to the lipid raft model. Lingwood and Simons (Lingwood and Simons, 2010) argue that proteins segregate into domains of preferred lipid phase, ordered or disordered. Once such segregation has occurred, would the association force described in this paper effectively disappear? Can the two processes, segregation into domains of preferred phases (at coexistence), and attraction between proteins within mismatched phases, be reconciled (or are they the equivalent)? Could the effect presented here be a driver for raft formation? In such a process, "mismatched" proteins in a membrane close to phase separation would aggregate into clusters, and entropic effects would then push the phase boundary out.

4) Molecular determinants. What would make a protein orderphilic, beyond having low hydrophobic mismatch with the ordered phase? Are there ways to modulate the strength and the range of the interaction?

5) Membrane remodeling. The authors briefly discuss a possible role in membrane fusion. Interestingly, lipid phase separation has been suggested to play a central role in ESCRT-protein induced coat-free vesicle budding (Rozycki et al., PLoS Comp Biol 8, e1002736, 2012).

6) It would be instructive to include the definition of h(x) in Figure 1.

7) The use of the word 'ordered' in both the *Abstract* and *Introduction* is misleading in the context of biomembranes. What the authors probably mean is solid-ordered, or gel, and not liquid-ordered. This distinction should be made from the beginning to avoid misunderstanding.

8) The final paragraph on the possible importance of the "order-phobic" force should be revised in light of the above comments.

[Editors' note: further revisions were requested prior to acceptance, as described below.]

Thank you for resubmitting your work entitled "Pre-transition effects mediate forces of assembly between transmembrane proteins" for further consideration at *eLife*. Your revised article has been favorably evaluated by Richard Aldrich (Senior editor), a Reviewing editor, and two reviewers. The manuscript has been improved but there are some minor remaining issues that need to be addressed before acceptance, as outlined below.

*Reviewer #2:*

The authors have largely addressed my concerns. Introduction *Introduction* and sections establish the biological relevance more clearly, without hiding the simplifications of the simulation model compared to biological membranes. Even if the new simulations of orderphilic proteins in a liquid-disordered membrane are only at a preliminary stage, the observed behavior clearly mirrors that of orderphobic proteins in a solid-ordered membrane, as would be expected. In my opinion, the new data strengthen the paper considerably, and I thus suggest including them.

Despite the simplifications compared to real biological membranes, the work provides strong evidence that the perturbation of lipid phase behavior by integral membrane proteins can create substantial driving forces for assembly. The paper should attract attention also by the experimental community and stimulate further explorations of the role of phase behavior. I recommend publication in *eLife*.

*Reviewer #3:*

My main previous concern, the biological relevance of the findings, is now much more clearly discussed in the revised paper.

There are two remaining aspects that still require some discussion:

1) The authors write "The orderphobic effect should be a general consequence of a first-order transition, whether the transition is between solid-ordered and liquid-disordered phases as considered explicitly herein, or between liquid-ordered and liquid-disordered phases as in multicomponent membrane systems". I strongly doubt that the transition between liquid-ordered and liquid-disordered phases is a first order transition. The experimental work of Veatch, Keller and co-workers (e.g. Veatch et al., ACS Chem Biol, 2008; Honerkamp-Smith et al., BBA Biomem, 2009), clearly shows that, upon cooling of a lipid extract (either from real plasma membranes or model membranes), the system shows critical behavior. I urge the authors to cite this work and discuss the implications thereof.

2) The authors should discuss the connection between their work and the work of Schäfer et al. (Shäfer et al., 2011) in more detail. The simulation studies of Schafer et al., based on the same (Martini) model that is used here, demonstrate that proteins are expelled from liquid-ordered domains as a result of 'orderphobicity'. Although the term orderphobic is not used, the driving forces for the partitioning of proteins into disordered domains are shown to be a direct consequence of the protein-induced perturbation of order in the liquid-ordered domains. In a subsequent study (Domanski et al., BBA Biomem, 2012), it is actually shown that these driving forces can lead to protein-induced domain formation. In the context of the biological significance of the current work, these studies should be properly discussed.

---

## [Author Response]

The primary concern of the reviewers appears to be the biological relevance of the solid-ordered phase used to demonstrate the orderphobic effect. We agree with the reviewers that the solid ordered phase has not been directly observed in most biological membranes (except lung surfactants). And we agree that our writing was inaccurate on this issue. Our revision emphasizes that biological membranes exhibit order–disorder phenomena in that significant portions exist in the liquid-ordered state, and these parts coexist with the liquid-disordered state. The liquid-ordered state is multicomponent and much more complicated than the one-component membrane considered in the current work. Nevertheless, our one-component system also exhibits an order–disorder transition between a solid-ordered phase and a liquid-disordered phase, which allows us to demonstrate the generic role of order–disorder in organizing proteins in what we think is the simplest possible context.

*1) The authors mention in the Introduction a number of papers (Nishimura et al., 2006; Polozov et al., 2008; Swamy et al., 2006; Munro, 2003; Thewalt and Bloom, 1992; Owen et al., 2012) that supposedly provide evidence for the physiological relevance of gel domains. Either the referenced papers deal with situations in which membranes are being cholesterol depleted or thermally quenched, or they talk about liquid-ordered domains, not gel domains. In fact, in Munro, 2003 it literally states "The solid gel phase is not thought to be of physiological relevance". Please change the Introduction accordingly.*

We agree that the original references (Nishimura et al., 2006; Polozov et al., 2008; Swamy et al., 2006; Munro, 2003; Thewalt and Bloom, 1992; Owen et al., 2012) did not provide evidence for the physiological relevance of solid domains and that the *Introduction* and *Implications* sections required revision. The comment is correct in that some references (Nishimura et al., 2006; Polozov et al., 2008; Swamy et al., 2006; Munro, 2003; Thewalt and Bloom, 1992; Owen et al., 2012) in our initial submission were about membrane temperature changes and cholesterol depletion. Nevertheless, those references are not irrelevant because such experiments provide important clues to understanding the ﬁnal state of biological membranes at physiologically relevant temperatures and compositions. That said, the original reference (Munro, 2003) was improperly placed, and we have changed its position now to be a reference for the lipid raft hypothesis.

We revised our *Introduction* to make our motivation clear as to why we studied the transition between solid-ordered and liquid-disordered phases to facilitate the simplest possible illustration of the generic orderphobic eﬀect. The importance of studying the ordered phase and the orderphobic eﬀect is further elaborated in the revised *Implications* section. Studying the ordered phases and the eﬀects of order–disorder transitions are also of direct relevance in understanding the behavior of lung-surfactant monolayers. The implications of the orderphobic eﬀect to monolayers is now added in our revised *Implications* section.

*2) The demonstration of the attraction between proteins in this manuscript concentrates on proteins that induce local disorder in an otherwise ordered phase. In biological membranes the disordered phase is expected to dominate, to ensure fluidity and facile transport. The authors may thus want to discuss the reversed situation in a bit more detail (which should be quite symmetric). Ideally, the authors are encouraged to consider a simulation that is more relevant for the biological situation.*

First, we direct the reviewers to our revised *Introduction* and *Implications* sections that discuss the importance of liquid-ordered phases in biological membranes, in addition to the disordered phase.

Second, concerning the dual to the orderphobic eﬀect, the orderphilic eﬀect, we have followed up on the suggestion of reviewers by reversing the roles of solid-ordered and liquid-disordered phases. Preliminary results of these calculations are shown in Figure 9,Figure 10. Bear in mind that without the additional components needed to produce a liquid-ordered phase, these results are no more or less relevant than those already studied for the orderphobic eﬀect.

In particular, with the disordered bilayer, we embedded a model protein with a hydrophobic thickness approximately equal to that of the ordered phase, ℓ=3.3 nm. This embedding illustrates the orderphilic case. As a control, we also embedded a model protein with a hydrophobic thickness approximately equal to that of the disordered phase, ℓ=2.3 nm. For both cases, we then calculated the average of the bond orientational order density ⟨ϕ(r)⟩ <)>, and the number density ⟨ρ(r)⟩ as deﬁned in the main text. We have also computed the director density, ⟨u(r)⟩, whereu(r)=∑lulδ(rl−r), with *_l_* = (3/2) cos*_l_*ul=(3/2)cos2⁡(θl)−(1/2). Here, is the angle between the membrane normal and the llth chain’s orientation. Further, with the procedure described in the text, we have computed the instantaneous interfaces from the coarse graining of each of these three ﬁelds ρ(r), and u(r). In the orderphilic case, interfaces are apparent for all three ﬁelds. The director ﬁeld provides the clearest pictures, which are illustrated in Figure 9,Figure 10. In the control case, interfaces do not appear.

Distinctions between interfaces for each of the ﬁelds are interesting microscopic eﬀects worthy of future study. For the bulk interface between the ordered and disordered phases, the distinction disappears because there is only one order–disorder transition for the membrane model we have considered. In more complicated membrane models, those with mixtures of components exhibiting both liquid-ordered and solid-ordered phases, the diﬀerent ﬁelds oﬀer diﬀerent information for both large and small length scales, as now mentioned in the *Implications* section and elsewhere.

*3) The authors may want to discuss in more depth the relation to the lipid raft model. Lingwood and Simons (Lingwood and Simons, 2010) argue that proteins segregate into domains of preferred lipid phase, ordered or disordered. Once such segregation has occurred, would the association force described in this paper effectively disappear? Can the two processes, segregation into domains of preferred phases (at coexistence), and attraction between proteins within mismatched phases, be reconciled (or are they the equivalent)? Could the effect presented here be a driver for raft formation? In such a process, "mismatched" proteins in a membrane close to phase separation would aggregate into clusters, and entropic effects would then push the phase boundary out.*

The comment raises interesting questions, but answering those questions is beyond the scope of what we have thus far done. We have considered the eﬀects of one or two proteins in a one-component membrane, while understanding lipid rafts and the like requires studies of the collective phase behavior of the mixtures of lipids and proteins including the orderphobic and orderphilic eﬀects. Appropriate modeling is possible, as the revised manuscript indicates, but it will involve extensive new work. The revised manuscript foreshadows studies of these questions in the *Implications* section.

*4) Molecular determinants. What would make a protein orderphilic, beyond having low hydrophobic mismatch with the ordered phase? Are there ways to modulate the strength and the range of the interaction?*

As already discussed in the earlier version of our paper, the strength and range of the interaction can be modulated by (a) proximity to the order–disorder transition (in temperature as well as surface pressure), (b) radial size of the solute, and (c) relative interactions between the lipid tail groups and the hydrophilic ‘caps’ of the protein. See the subsection “The orderphobic eﬀect generates forces of assembly and facilitates protein Mobility” of the revised version. An additional factor is that changing components of the membrane change the proximity to the order–disorder transition, thus also aﬀecting the strength and range of the eﬀect. This last aspect is added to the *Implications* section of the revised manuscript.

Note also that the Appendix shows how a model *α*-helix is orderphobic even though the chosen peptide has low hydrophobic mismatch with the ordered phase. This ﬁnding demonstrates that side-chains of a protein are suﬃcient to perturb the ordered structure of the membrane, making *α*-helices orderphobic. Similarly, it should be possible to make a protein less orderphobic by altering its side-chain chemistry, or secondary structure (e.g., *β*-sheet). A detailed characterization of how proteins of diﬀerent hydrophobic mismatches, secondary structures, and side-chain chemistries modulate the orderphobic eﬀect is left to future work. This possibility is noted in the *Implications* section.

*5) Membrane remodeling. The authors briefly discuss a possible role in membrane fusion. Interestingly, lipid phase separation has been suggested to play a central role in ESCRT-protein induced coat-free vesicle budding (Rozycki et al., PLoS Comp Biol 8, e1002736, 2012).*

We have changed the manuscript to include this reference.

*6) It would be instructive to include the definition of h(x) in Figure 1.*

We have modiﬁed Figure 1 to include an illustration of h(x).

*7) The use of the word 'ordered' in both the Abstract and Introduction is misleading in the context of biomembranes. What the authors probably mean is solid-ordered, or gel, and not liquid-ordered. This distinction should be made from the beginning to avoid misunderstanding.*

We have revised the *Introduction* accordingly.

*8) The final paragraph on the possible importance of the 'orderphobic' force should be revised in light of the above comments.*

We have revised the *Introduction* and *Implications* sections.

[Editors' note: further revisions were requested prior to acceptance, as described below.]

*Reviewer #2: The authors have largely addressed my concerns. Introduction and Implications sections establish the biological relevance more clearly, without hiding the simplifications of the simulation model compared to biological membranes. Even if the new simulations of orderphilic proteins in a liquid-disordered membrane are only at a preliminary stage, the observed behavior clearly mirrors that of orderphobic proteins in a solid-ordered membrane, as would be expected. In my opinion, the new data strengthen the paper considerably, and I thus suggest including them. Despite the simplifications compared to real biological membranes, the work provides strong evidence that the perturbation of lipid phase behavior by integral membrane proteins can create substantial driving forces for assembly. The paper should attract attention also by the experimental community and stimulate further explorations of the role of phase behavior. I recommend publication in eLife.*

We are pleased with the reviewer’s assessment. We have followed his/her advice and added a section to the Appendix in which we show preliminary results for the orderphilic case. Also, in the *Implications* section, we have added a few more words both to point to the Appendix and to explain the relevance to lipid rafts.

*Reviewer #3: My main previous concern, the biological relevance of the findings, is now much more clearly discussed in the revised paper. There are two remaining aspects that still require some discussion: 1) The authors write "The orderphobic effect should be a general consequence of a first-order transition, whether the transition is between solid-ordered and liquid-disordered phases as considered explicitly herein, or between liquid-ordered and liquid-disordered phases as in multicomponent membrane systems". I strongly doubt that the transition between liquid-ordered and liquid-disordered phases is a first order transition. The experimental work of Veatch, Keller and co-workers (e.g. Veatch et al., ACS Chem Biol, 2008; Honerkamp-Smith et al., BBA Biomem, 2009), clearly shows that, upon cooling of a lipid extract (either from real plasma membranes or model membranes), the system shows critical behavior. I urge the authors to cite this work and discuss the implications thereof.*

We appreciate the reviewer’s thoughts on this issue, but with all due respect, we disagree with his/her impression that the first-order transition is not present in these systems. It is true that, for a fixed cholesterol concentration, there is a critical temperature associated with the transition, as located by Veatch and Keller, but below that temperature, there is two-phase coexistence, and the transition at those lower temperatures is first-order. (The same is true for a liquid–vapor transition: below its critical temperature, there is two-phase coexistence, and at such temperatures the transition between those phases is first order.) It is a general principle that at the conditions where two phases with different order-parameter values coexist, the transition between the phases is first order. The reference to Veatch and Keller that we do supply in the *Implications* section of the paper establishes the region of two-phase coexistence between liquid-ordered and liquid-disordered phases. The temperatures and lipid concentrations for which this coexistence occurs depends upon additional parameters, such as cholesterol concentrations and membrane surface tension.

Because there is some confusion over this point, we have added further explanation concerning first-order and coexistence to the *Implications* section. We have also added a sentence that contrasts the orderphobic and orderphilic effects, which occur broadly for any membrane close to order–disorder coexistence, with the Casimir effect, which occurs over the much narrower conditions of order–disorder criticality.

*2) The authors should discuss the connection between their work and the work of Schäfer et al. (Shäfer et al. 2011) in more detail. The simulation studies of Schafer et al., based on the same (Martini) model that is used here, demonstrate that proteins are expelled from liquid-ordered domains as a result of 'orderphobicity'. Although the term orderphobic is not used, the driving forces for the partitioning of proteins into disordered domains are shown to be a direct consequence of the protein-induced perturbation of order in the liquid-ordered domains. In a subsequent study (Domanski et al., BBA Biomem, 2012), it is actually shown that these driving forces can lead to protein-induced domain formation. In the context of the biological significance of the current work, these studies should be properly discussed.*

In the *Implications* section, we have added discussion of the results that the reviewer points us to.